# Suppression of Linear Ubiquitination Ameliorates Cytoplasmic Aggregation of Truncated TDP-43

**DOI:** 10.3390/cells11152398

**Published:** 2022-08-03

**Authors:** Qiang Zhang, Seigo Terawaki, Daisuke Oikawa, Yoshinori Okina, Yoshinosuke Usuki, Hidefumi Ito, Fuminori Tokunaga

**Affiliations:** 1Department of Pathobiochemistry, Graduate School of Medicine, Osaka City University, Osaka 545-8585, Japan; d19ma301@st.osaka-cu.ac.jp (Q.Z.); terawaki@med.kawasaki-m.ac.jp (S.T.); oikawa.daisuke@omu.ac.jp (D.O.); 2Department of Molecular and Genetic Medicine, Kawasaki Medical School, Kurashiki 701-0192, Japan; 3Department of Medical Biochemistry, Graduate School of Medicine, Osaka Metropolitan University, Osaka 545-8585, Japan; okina.yoshinori@omu.ac.jp; 4Department of Chemistry, Graduate School of Science, Osaka Metropolitan University, Osaka 558-8585, Japan; usuki@omu.ac.jp; 5Department of Neurology, Wakayama Medical University, Wakayama 641-8510, Japan; itohid@kuhp.kyoto-u.ac.jp

**Keywords:** ALS, cytoplasmic aggregation, LUBAC, NF-κB, TDP-43, ubiquitin

## Abstract

TAR DNA-binding protein 43 (TDP-43) is a predominant component of inclusions in the brains and spines of patients with amyotrophic lateral sclerosis (ALS). The progressive accumulation of inclusions leads to proteinopathy in neurons. We have previously shown that Met1(M1)-linked linear ubiquitin, which is specifically generated by the linear ubiquitin chain assembly complex (LUBAC), is colocalized with TDP-43 inclusions in neurons from *optineurin*-associated familial and sporadic ALS patients, and affects NF-κB activation and apoptosis. To examine the effects of LUBAC-mediated linear ubiquitination on TDP-43 proteinopathies, we performed cell biological analyses using full-length and truncated forms of the ALS-associated Ala315→Thr (A315T) mutant of TDP-43 in Neuro2a cells. The truncated A315T mutants of TDP-43, which lack a nuclear localization signal, efficiently generated cytoplasmic aggregates that were colocalized with multiple ubiquitin chains such as M1-, Lys(K)48-, and K63-chains. Genetic ablation of *HOIP* or treatment with a LUBAC inhibitor, HOIPIN-8, suppressed the cytoplasmic aggregation of A315T mutants of TDP-43. Moreover, the enhanced TNF-α-mediated NF-κB activity by truncated TDP-43 mutants was eliminated in the presence of HOIPIN-8. These results suggest that multiple ubiquitinations of TDP-43 including M1-ubiquitin affect protein aggregation and inflammatory responses in vitro, and therefore, LUBAC inhibition ameliorates TDP-43 proteinopathy.

## 1. Introduction

Amyotrophic lateral sclerosis (ALS) is a fatal neurodegenerative disease characterized by the progressive degeneration of motor neurons in the brain and spinal cord [1,2]. Although most ALS cases are sporadic (sALS), ~10% of ALS patients are familial (fALS) and over 30 potential ALS genes have been identified in fALS patients. These gene products seem to regulate multiple cellular functions such as RNA metabolism and proteostasis, protein trafficking, ubiquitin-proteasomal, and autophagic degradation, cytoskeletal and axonal dynamics, and neuroinflammation. *TARDBP* encodes TAR DNA-binding protein 43 (TDP-43), which is a key component of the insoluble and ubiquitinated cytoplasmic inclusions in the brain and spinal cord of ALS and frontotemporal dementia (FTD) patients [3]. TDP-43 contains two RNA recognition motifs (RRM1 and RRM2) at the N-terminal region, which bind RNA/DNA, and a prion-like Gln/Asn-rich domain and a Gly-rich region at the C-terminal portion [4]. TDP-43 also contains a nuclear localization signal (NLS) and a nuclear export signal (NES), suggesting that it functions as a nucleocytoplasmic shuttling protein. TDP-43-positive cytoplasmic inclusions are detected in most ALS cases, and are considered to decrease RNA metabolism in the nucleus and/or increase proteotoxicity in the cytoplasm. Importantly, TDP-43 is susceptible to several intracellular proteases such as caspases, calpain, and asparaginyl endopeptidase, resulting in the generation of its C-terminal fragments (CTFs) such as TDP-35 and TDP-25 [5,6,7]. This proteolysis facilitates the cytoplasmic accumulation of insoluble CTFs [8,9], since CTFs are intrinsically disordered low-complexity domains that form cross-β amyloid structures [10]. Indeed, the deletion or mutations of the NLS facilitate the cytoplasmic localization and aggregation of TDP-43 [11,12]. These pathological aggregates contribute to cellular dysfunction and toxicity, which are collectively referred to as proteinopathy [3]. Moreover, various post-translational modifications such as ubiquitination, phosphorylation, acetylation, oxidation, SUMOylation, and PARylation also affect the proteinopathy of TDP-43 [3,13,14]. 

Protein ubiquitination is catalyzed by ubiquitin-activating enzyme (E1), ubiquitin-conjugating enzyme (E2), and ubiquitin ligase (E3), and regulates numerous cellular functions by generating various types of ubiquitin linkages in the “ubiquitin code” [15]. Typically, ubiquitin forms chains via isopeptide bonds between seven internal lysines (K6, K11, K27, K29, K33, K48, and K63). Among them, the K48-linked ubiquitination induces proteasomal degradation, whereas the K63-linked chain participates in nonproteolytic cellular functions such as membrane trafficking, DNA repair, and signal transduction. Homotypic polyubiquitin chains as well as branched and hybrid chains composed of multiple ubiquitin chains regulate various cellular functions [16]. The linear ubiquitin chain assembly complex (LUBAC), composed of the HOIL-1L (also known as RBCK1), HOIP (RNF31), and SHARPIN subunits, specifically generates a unique Met1(M1)-linked linear polyubiquitin chain [17], and is required for the regulation of the canonical nuclear factor-κB (NF-κB) activation and cell death pathways [18]. 

We previously reported that optineurin (OPTN) selectively binds to linear ubiquitin through the ubiquitin binding in ABIN and NEMO (UBAN) domain, which was initially identified as a linear ubiquitin-specific binding domain in NF-κB-essential modulator (NEMO) [19], and plays a crucial role in the suppression of NF-κB activity [20]. Furthermore, the fALS-associated *OPTN* mutations such as E478G and Q398X abrogated the inhibitory effects of OPTN on LUBAC-mediated NF-κB activation. Notably, M1-ubiquitin and activated NF-κB are colocalized with TDP-43 inclusions in neurons from *OPTN*-associated fALS patients with increased caspase activation [20,21]. We also showed that M1- and K63-ubiquitins are exclusively colocalized with thick bundles of TDP-43 from sALS patients [22] and tau neurofibrillary tangles from Alzheimer’s disease patients [23], whereas K48-ubiquitin was detected in both tiny and thick inclusions. These results suggest that protein aggregates such as TDP-43 and tau are initially conjugated with the proteasomal degradation-associated K48-chain and then undergo complex ubiquitination including non-degradative M1- and/or K63-ubiquitin chains, which may affect the proteinopathies of neurodegeneration diseases. 

We previously developed α,β-unsaturated carbonyl-containing LUBAC inhibitors, named HOIPINs, and determined that the compound named HOIPIN-8 was the most potent and specific inhibitor of LUBAC [24,25,26]. HOIPIN-8 suppresses the LUBAC-mediated proinflammatory cytokine-induced NF-κB activation by modifying the active Cys885 in the RING2 domain of HOIP, and attenuates the enhanced NF-κB activity in *OPTN*-deficient cells [26]. To investigate the progressive generation of complex ubiquitin chains in ALS proteinopathy, in this study, we expressed TDP-43 and its fragments in Neuro2a cells, and analyzed their aggregate formation and the effects of LUBAC suppression in vitro. 

## 2. Materials and Methods

### 2.1. Reagents

HOIPIN-8 (2-{(*E*)-3-[2,6-difluoro-4-(1*H*-pyrazol-4-yl)-phenyl]-3-oxo-propenyl}-4-(1-methyl-1*H*-pyraol-4-yl)-benzoic acid sodium salt) was synthesized as described [25]. The following reagents were obtained as indicated: recombinant mouse TNF-α (BioLegend, San Diego, CA, USA), MG-132 (Peptide Institute Inc., Osaka, Japan), chloroquine sulfate (Sigma-Aldrich, St. Louis, MO, USA), and DAPI (Dojindo, Kumamoto, Japan). 

### 2.2. Plasmids

The plasmids containing cDNAs of human wild type (WT) and the ALS-related Ala315→Thr (A315T) mutant of TDP-43 were kind gifts from Prof. Yoshitaka Nagai (Kindai University, Osaka-Sayama, Japan). The coding sequences were tagged with GFP, amplified by PCR, and cloned into the pcDNA3.1 expression vector (Invitrogen, Carlsbad, CA, USA). The open reading frames of the mouse and human LUBAC subunit cDNAs were amplified by reverse-transcription PCR. The truncation and deletion mutants of TDP-43 cDNAs were prepared by PCR, and the nucleotide sequences were verified. The cDNAs were ligated to the appropriate epitope sequences and cloned into the pcDNA3.1 or pCSII-EF-IRES2 vector (RIKEN BioResource Research Center, Tsukuba, Japan).

### 2.3. Cell Culture and Transfection

Murine neuroblastoma Neuro2a cells (ATCC, Manassas, VA, USA) were cultured in MEM (Nacalai Tesque, Kyoto, Japan) containing 10% fetal bovine serum (FBS), supplemented with non-essential amino acids and penicillin/streptomycin (Nacalai Tesque). HEK293T cells (ATCC) were cultured in DMEM, containing 10% fetal bovine serum, 100 IU/mL penicillin G, and 100 μg/mL streptomycin, at 37 °C under a 5% CO_2_ atmosphere. Transfections were performed using PEI Max (Polysciences, Warrington, PA, USA) or Lipofectamine 2000 (Thermo Fisher, Waltham, MA, USA). To establish tetracycline-inducible TDP43-GFP expression cell lines, Neuro2a cells were sequentially transfected with pTet-On-Zeo, pTRE2-Hyg containing GFP-tagged TDP43 cDNA variants, and mKate2/pcDNA3.1, and then selected with 100 μg/mL zeocin (Invivogen, San Diego, CA, USA), 300 μg/mL hygromycin, and 400 μg/mL G418 (Nacalai Tesque), respectively. The number of viable cells was measured with a CellTiter-Glo Luminescent Cell Viability Assay (Promega, Madison, WI, USA), which quantifies the ATP content.

### 2.4. Immunofluorescent Staining and Imaging Analyses

Neuro2a cells (4 × 10^4^ cells) were seeded on poly-D-lysine-coated coverslips in 24-well plates, one day before transfection. The plasmids encoding TDP-43 and/or LUBAC were transfected into Neuro2a cells, using PEI. After 24 h, the cells were fixed with phosphate buffered 4% paraformaldehyde at room temperature for 15 min, and then permeabilized/blocked in staining buffer (0.05 % saponin, 10 % FBS, 10 mM glycine in PBS) for 30 min. The cells were incubated overnight with primary antibodies at 4 °C in a humidity box. The next day, the cells were washed, incubated with secondary antibodies and DAPI, and then mounted onto glass slides with FluorSave (MilliporeSigma, Burlington, MA, USA). The confocal fluorescence images of the prepared slides were captured with an LSM700 or LSM800 confocal microscope system (Carl Zeiss, Oberkochen, Germany). All images were acquired as 16-bit depth images with a 20× dry, 40× oil-, or 63× water-immersion objective lens by scanning each channel two separate times for averaging. In quantitative imaging analyses using the Tet-On inducible TDP-43 expressing cells, fluorescence images of TDP-43-expressing cells in 24-well plates were randomly acquired by an IN Cell Analyzer 2500HS with a 60× objective lens (GE Healthcare, Chicago, IL, USA). Fluorescence intensities and sizes of TDP-43 aggregates in the images were analyzed by Cell Profiler ver. 4.2.1 (Broad Institute Inc., Cambridge, MA, USA) [27]. 

### 2.5. Cell Lysis, Immunoprecipitation, SDS-PAGE, and Immunoblotting

Detergent-soluble and -insoluble protein fractions were prepared by stepwise extraction. First, cells were lysed with RIPA buffer (50 mM Tris-HCl, pH 7.4, 150 mM NaCl, 1% (*w*/*v*) NP-40, 0.5% (*w*/*v*) sodium deoxycholate, 0.1% (*w*/*v*) SDS, and 2 mM EDTA) supplemented with 2 mM PMSF and protease inhibitor cocktail (Sigma-Aldrich). After centrifugation, the pellets were washed with RIPA buffer and then solubilized with urea lysis buffer (50 mM Tris-HCl, pH 7.5, 6 M urea, 150 mM NaCl, and 1% Triton X-100). For immunoprecipitation analyses, cells were lysed with 50 mM Tris-HCl, pH 7.5, 150 mM NaCl, 1% Triton X-100, 2 mM PMSF, and complete protease inhibitor cocktail (Sigma-Aldrich). To prepare completely denatured cell lysates, the cells were solubilized in 50 mM Tris-HCl, pH 7.5, 150 mM NaCl, 1.2% SDS, and 5% sucrose, and then heated for 15 min at 95 °C. The hot-SDS lysates were centrifuged, and the supernatants were diluted 8-fold with 50 mM Tris-HCl, pH 7.5, containing 150 mM NaCl and 0.1% NP-40. Immunoprecipitation was performed using appropriate antibodies, followed by Protein G agarose beads (GE Healthcare) at 4 °C with gentle rotation. Immunoprecipitates were washed five times with 50 mM Tris-HCl, pH 7.5, 6 M urea, 150 mM NaCl, and 1% Triton X-100. All types of samples were then separated by SDS-PAGE and transferred to the PVDF membranes. After blocking in Tris-buffered saline containing 0.1% Tween-20 (TBS-T) with 5% skim-milk, the membrane was incubated with the appropriate primary antibodies, and then with horseradish peroxidase-conjugated secondary antibodies (GE Healthcare). The chemiluminescent images were obtained and the intensities were quantified with an LAS4000 imaging analyzer (GE Healthcare) or a Fusion Solo S imaging system (Vilber, Collégien, France).

### 2.6. Construction of Hoip-Knockout Neuro2a Cells

The 5′-AGGAGCTGGCGAGCGCCCTGAGG-3′ nucleotide sequences in exon 1 of the mouse *Hoip* gene were selected as the targets. Double-stranded target DNAs were prepared by annealing synthesized oligo DNAs, and then incorporated into the *Bbs*I site of the px458 CRISPR/Cas9 vector (Addgene, Watertown, MA, USA). Each plasmid was transiently introduced into Neuro2a cells, and 2 days after transfection, the GFP positive cells were sorted by a FACSAria IIIu (BD Biosciences, San Jose, CA, USA). Single clones were obtained by limiting dilution. Genome editing of the *Hoip* gene was confirmed by genome sequencing and immunoblotting.

### 2.7. Antibodies

The following antibodies were used for immunoblot analyses: TDP-43 (Invitrogen, JM51-10, 1:2000), HOIP (Abcam, Cambridge, UK, ab125189, 1:1000), M1-ubiquitin (Genentech, South San Francisco, CA, USA, 1F11/3F5/Y102L, 50 ng/mL), K48-ubiquitin (MilliporeSigma, Apu2, 1:1000), K63-ubiquitin (MilliporeSigma, Apu3, 1:1000), GFP (Clontech Laboratories, Mountain View, CA, USA, JL-8, 1:2000), HA (Roche, Basel, Switzerland, 11867423001, 1:1000), Myc (Santa Cruz Biotechnology, Dallas, TX, USA, 9E10, 1:1000 and MBL, Tokyo, Japan, HRP-Conjugate, M192-7; 1:20,000), and DYKDDDDK-tag (Merck, Darmstadt, Germany, M2, 1:2000 and Wako, Osaka, Japan, 1E6, 015-22391, HRP-Conjugate; 1:20,000). For immunoprecipitation, FLAG (Sigma-Aldrich, clone M2, F1840, 1 μg) and normal rabbit IgG (MBL, PM035, 1 μg) were used. The primary antibodies TDP-43 (Invitrogen, JM51-10, 1:500), M1-ubiquitin (Genentech, 1F11/3F5/Y102L, 1 μg/mL), ubiquitin (MBL, FK2, 1:2000), K48-ubiquitin (MilliporeSigma, Apu2, 1:500), and K63-ubiquitin (MilliporeSigma, Apu3, 1:100) were used for immunofluorescence analyses, followed by Alexa Fluor 488, 546, or 647 labeled anti-mouse, anti-human, or anti-rabbit IgG (Thermo Fisher, goat polyclonal; 1:1000) as secondary antibodies.

### 2.8. Luciferase Assay 

Neuro2a cells (5 × 10^4^ cells/well) were cultured in 24-well plates overnight, and then co-transfected with the TDP-43 expression vector, the LUBAC expression vector, the pGL4.32 [*luc2P*/NF-κB-RE/Hygro] vector (Promega), and the pRL-TK *Renilla* Luciferase control reporter vector (Promega) using Lipofectamine 2000. The medium was replaced at 4 h-post transfection. At 24 h after transfection, the cells were lysed and the luciferase activity was measured with a GloMax 20/20 luminometer (Promega) using the Dual-Luciferase Reporter Assay System (Promega). 

### 2.9. Statistics

The Mann–Whitney U test, Kruskal–Wallis test, *t*-test, and one-way ANOVA followed by a post hoc Tukey HSD test were performed using the GraphPad Prism 8 software. For all tests, a *p* value of less than 0.05 was considered statistically significant.

## 3. Results

### 3.1. Expression of TDP-43 Truncations Generates Multiple Ubiquitin Chain-Positive Cytoplasmic Aggregates

To investigate the involvement of M1-linked linear ubiquitination in TDP-43 proteinopathy, we first constructed GFP-tagged full-length and truncated forms of the ALS-associated A315T mutant of TDP-43 (Figure 1A), and ectopically expressed them in mouse neuroblastoma Neuro2a cells. Immunofluorescence analyses demonstrated that the GFP alone was expressed throughout the cell, and the A315T^1-414^-GFP was localized predominantly in the nucleus without aggregation. In contrast, the truncated mutants of TDP-43 such as A315T^89-414^-GFP, A315T^247-414^-GFP, and A315T^1-414^-ΔNLS-GFP efficiently formed cytoplasmic aggregates, which were colocalized with the pan- and M1-linked ubiquitins (Figure 1B and Appendix A), suggesting that TDP-43 mutants lacking the NLS generated linear ubiquitin-positive cytoplasmic aggregates in vitro.

To further investigate the linkages of the ubiquitin chains in the cytoplasmic TDP-43 aggregates, we next examined the colocalization of M1- and proteasomal degradation-inducible K48-linked ubiquitin chains (Figure 2A and Appendix A). The M1-ubiquitin-positive cytoplasmic aggregates of A315T^89-414^-GFP and A315T^247-414^-GFP, but not GFP and A315T^1-414^-GFP, also reacted with the anti-K48-ubiquitin antibody, suggesting that the cytoplasmic aggregates of truncated TDP-43 include both M1- and K48-ubiquitin chains. In the presence of the proteasomal inhibitor, MG-132, the formation of cytoplasmic aggregates of TDP-35 and TDP-25 was reportedly upregulated and stabilized [7]. Therefore, to examine the effects of inhibitors of proteasomal or lysosomal/autophagic degradation on TDP-43 aggregate formation, we treated A315T-GFPs-expressing Neuro2a cells with MG-132 or chloroquine. The cells were lysed with RIPA buffer to obtain the detergent-soluble fraction, and the resultant pellet was further solubilized with 6 M urea-containing buffer to obtain the RIPA-insoluble/urea-soluble aggregate fraction. As shown in Figure 2B, the number of truncated forms of A315T-GFPs in the insoluble aggregate fraction were increased in the presence of MG-132, but not chloroquine. Moreover, A315T^247-414^-GFP migrated as lower molecular weight bands in both the RIPA-soluble and urea-soluble fractions in the absence or presence of MG-132 and chloroquine, indicating its susceptibility to endogenous proteolytic enzymes. A smeared band, representing the putative polyubiquitination of A315T^247-414^-GFP, was detected in the RIPA-insoluble and urea-soluble fraction. These results suggest that the inhibition of proteasome activity facilitates the aggregation of A315T-GFPs. In MG-132-treated Neuro2a cells, the cytoplasmic aggregates of A315T^89-414^-GFP and A315T^247-414^-GFP were increased, and the colocalization of M1- and pan-ubiquitins (Figure 2C) as well as the M1- and K48-ubiquitins (Figure 2D) with the A315T-GFP truncated aggregates was detected. 

Similarly, the colocalization of the K63- and M1-ubiquitins was detected in the cytoplasmic aggregates of A315T^89-414^-GFP and A315T^247-414^-GFP, but not GFP and A315T^1-414^-GFP (Figure 3A). In the presence of MG-132, the cytoplasmic aggregates of A315T^89-414^-GFP and A315T^247-414^-GFP colocalized with K63- and M1-ubiquitins were increased (Figure 3B). Collectively, these results suggest that multiple ubiquitin chains such as M1-, K48-, and K63-ubiquitin chains are cooperatively conjugated to the cytoplasmic aggregates of truncated TDP-43 mutants, and the inhibition of proteasome activity, which basically affects the K48-linked ubiquitin chain, increases the number of cytoplasmic aggregates containing multiple ubiquitin chains. 

### 3.2. LUBAC Plays an Important Role in Cytoplasmic Aggregation of Truncated TDP-43 

To clarify the involvement of LUBAC activity in the formation of TDP-43 aggregates, we constructed *Hoip*-knockout (KO) Neuro2a cells by the CRISPR/Cas9 technique, resulting in the elimination of HOIP and reduced the HOIL-1L and SHARPIN levels (Appendix A). When full length A315T^1-414^-GFP was expressed in the *Hoip*-KO cells, nuclear localization and non-aggregate formation were detected, as in the parental Neuro2a cells (Figure 4A). In contrast, the number of cytosolic aggregates of A315T^247-414^-GFP were reduced in the *Hoip*-KO cells compared to those in the parental cells, and indeed, the intensity and size of the A315T^247-414^-GFP aggregates were significantly reduced in the *Hoip*-KO cells (Figure 4B,C). Furthermore, the amount of A315T^247-414^-GFP in the RIPA-insoluble/urea-soluble aggregate fraction was reduced in the absence of *Hoip*, although almost equal numbers of full length A315T^1-414^-GFP were detected in both the parental and *Hoip*-KO Neuro2a cells (Figure 4D). Importantly, M1-ubiquitin was detected in the insoluble fraction of the parental cells, but not the *Hoip*-KO cells. In the presence of MG-132, the accumulation of A315T^247-414^-GFP in the insoluble fraction was enhanced in both the parental and *Hoip*-KO Neuro2a cells, and the chain length of M1-ubiquitin seemed to be extended. 

To further examine the involvement of LUBAC activity, we constructed a LUBAC expression plasmid that could simultaneously express the HOIP, HOIL-1L, and SHARPIN subunits on a single plasmid (Appendix A). The restoration of the active LUBAC (LUBAC-WT) within the *Hoip*-KO cells recovered the M1-ubiquitin-positive aggregates of A315T^247-414^-GFP (Figure 4E). In contrast, the restoration of LUBAC-CA, in which the active Cys879 of mouse HOIP was replaced by Ala, did not generate M1-ubiquitin or A315T^247-414^-GFP aggregates. For biochemical confirmation, we performed the immunoblotting of A315T-GFPs in the RIPA-soluble and RIPA-insoluble/urea-soluble fractions (Figure 4F). Although the expression of A315T^1-414^-GFP in the RIPA-insoluble/urea-soluble fraction was not affected, regardless of the LUBAC activity in the parental and *Hoip*-KO Neuro2a cells, the amount of A315T^247-414^-GFP in the aggregate fraction was enhanced in the presence of LUBAC-WT, but not LUBAC-CA, in the parental and *Hoip*-KO Neuro2a cells. These results strongly indicate that the LUBAC-mediated linear ubiquitination activity is involved in the formation of cytoplasmic aggregates of the truncated TDP-43 mutant. Therefore, we hypothesized that the inhibition of LUBAC activity may suppress the aggregation of TDP-43.

### 3.3. Inhibition of LUBAC by HOIPIN-8 Reduces Cytoplasmic Aggregation of Truncated TDP-43

In the presence of the LUBAC inhibitor HOIPIN-8, LUBAC-mediated M1-polyubiquitination was strongly suppressed (Appendix A), as reported [25,26]. When we treated truncated WT^247-414^- or A315T^247-414^-GFP-expressing Neuro2a cells with HOIPIN-8, the number of cytoplasmic aggregates colocalized with M1- and pan-ubiquitins were decreased (Figure 5A). The anti-GFP immunoblotting of RIPA-insoluble/urea-soluble aggregate fractions of WT^247-414^-GFP- and A315T^247-414^-GFP-expressing Neuro2a cells revealed that the MG-132-treated cells contained larger amounts with smeared migrations of WT^247-414^-GFP and A315T^247-414^-GFP, with increased K48- and K63-ubiquitinations compared to nontreated cells (Figure 5B). Conversely, in the presence of HOIPIN-8, the amounts of insoluble WT^247-414^-GFP and A315T^247-414^-GFP were reduced either with or without MG-132. Although HOIPIN-8 reduced the amount of M1-ubiquitin in the insoluble fraction, it did not affect the amount of K48-ubiquitin in the insoluble fraction. Further analyses demonstrated that HOIPIN-8 dose-dependently suppressed the insoluble aggregation of A315T^247-414^-GFP (Figure 5C,D), and the quantitative analysis also showed that the intensity and size of the A315T^247-414^-GFP aggregates were significantly reduced by HOIPIN-8 (Figure 5E). These results suggest that LUBAC activity is involved in TDP-43 aggregation, and the suppression of LUBAC-mediated M1-ubiquitination by HOIPIN-8 ameliorates the aggregation of truncated TDP-43. 

### 3.4. LUBAC Associates with and Ubiquitinates Truncated TDP-43

To further examine the involvement of LUBAC in the TDP-43 aggregate formation, we overexpressed LUBAC and then performed an immunofluorescence analysis. The results demonstrated that HOIP, a catalytic subunit of LUBAC, was localized at the peripheries of the M1-ubiquitin and A315T^247-414^-GFP aggregates (Figure 6A). The immunoprecipitation analysis indicated that LUBAC, but not its respective subunit, was coprecipitated with A315T^247-414^-GFP in the HEK293T cells, whereas GFP alone did not bind LUBAC (Figure 6B). To determine whether LUBAC activity is necessary for interaction with A315T^247-414^-GFP, we performed a co-immunoprecipitation analysis using HOIP-WT or -CA with HOIL-1L and SHARPIN expressed in the HEK293T cells (Figure 6C). The anti-GFP antibody efficiently co-immunoprecipitated A315T^247-414^-GFP with HOIP-WT-containing active LUBAC, but not the inactive HOIP-CA-containing LUBAC, suggesting that LUBAC activity plays a crucial role in A315T^247-414^-GFP-binding. A scheme of the truncated TDP-43-LUBAC binding is shown in Figure 6D. To examine whether LUBAC is an E3 for TDP-43, we constructed the doxycycline (Dox)-inducible WT^247-414^- and A315T^247-414^-GFP-expressing Neuro2a cells (Appendix A). Although the overexpression of GFP-TDP-43-A315T in SH-SY5Y cells reportedly enhances apoptosis and autophagy [28], the induction of WT^247-414^- or A315T^247-414^-GFP did not result in a statistically significant decrease in cell viability (Appendix A). Immunoprecipitations of the hot-SDS Neuro2a cell lysates suggest that A315T^247-414^-GFP was covalently conjugated with M1-, K48-, and K63-ubiquitins after Dox-induction (Figure 6E). The HOIPIN-8 treatment reduced the M1-ubiquitination, but not the K48- nor K63-ubiquitination of A315T^247-414^-GFP, whereas the M1-, K48-, and K63-ubiquitinations of A315T^247-414^-GFP were all drastically increased in the presence of MG-132. In contrast, in the presence of both HOIPIN-8 and MG-132, the enhanced M1/K48/K63-ubiquitination of A315T^247-414^-GFP was suppressed. These results suggest that multiple ubiquitins are cooperatively conjugated to the truncated TDP-43, and HOIPIN-8 suppresses the progression toward forming a complex ubiquitin chain.

### 3.5. HOIPIN-8 Suppresses Truncated TDP-43-Induced Upregulation of NF-κB Activity 

Finally, to investigate the effects of TDP-43 and HOIPIN-8 on neuroinflammatory responses, we performed an NF-κB luciferase reporter assay in the TNF-α-treated Neuro2a and HEK293T cells (Figure 7 and Appendix A). Compared with GFP alone, WT^1-414^- or A315T^1-414^-GFP showed no effect on the TNF-α-induced NF-κB activity. In contrast, the TDP-43 truncation mutants, WT^247-414^- and A315T^247-414^-GFP, upregulated the NF-κB activity. In the HOIPIN-8-treated cells, the NF-κB activity was suppressed by the expression of GFP or any of the TDP-43-GFPs. These results suggest that the aggregate-prone TDP-43 truncation mutants potentially upregulate the NF-κB activity in the presence of an inflammatory cytokine, and that HOIPIN-8 cancels the increased neuroinflammatory responses.

## 4. Discussion

ALS is a fatal neurodegenerative disease. Analyses of fALS have revealed the causative genes of ALS such as *TARDBP* (which encodes TDP-43), *SOD1*, *FUS*, *C9ORF72*, *ATXN2*, *OPTN*, *VCP*, *UBQLN2*, *UBQLN4*, *SQSTM1*, *TBK1*, and so on, but the mechanism of ALS onset has remained elusive. TDP-43-containing cytoplasmic inclusions are reportedly detected in 97% of ALS and 45% of FTD cases [29]. Therefore, TDP-43 inclusions are a hallmark of ALS/FTD-mediated proteinopathy [3,4]. The recent structural analysis of TDP-43^282-360^ showed that the region has a unique double-spiral-shaped fold, and is oligomerized with a prion-like filamentous structure [10]. Ubiquitination as well as phosphorylation is a crucial post-translational modification of TDP-43 in the brain inclusions of ALS patients [11,30]. TDP-43 contains 20 Lys residues, and multiple ubiquitination sites have been identified [31,32]. Importantly, Scotter et al. showed that the TDP-43 aggregates included both the K48- and K63-linked ubiquitin chains, which function in proteasomal degradation and autophagic degradation, respectively [33]. Furthermore, Yin et al. reported the age-dependent increase in the K63-ubiquitination of the C-terminal fragment of TDP-43, with binding to proteasome assembly proteins, PSM2 and PSD13 [34]. We determined that not only K48- and K63-linked ubiquitin chains, but also M1-linked linear ubiquitin, which regulates NF-κB activation and the cell death pathway, are involved in inclusions from *OPTN*-associated fALS and sALS patients, and in tau neurofibrillary tangles from Alzheimer’s disease patients [20,22,23]. Although K48-ubiquitin was detectable in tiny inclusions, K63- and M1-positive inclusions were observed in K48-positive thick inclusions. The M1-ubiquitin was also colocalized with protein aggregates formed by the overexpression of pathogenic huntingtin-derived polyglutamine proteins and ataxin-3 (Machado–Joseph disease) [35]. These results suggest that the ubiquitin chains of neurodegenerative disease-associated inclusions become progressively more complex.

In this study, we showed that cytoplasmic aggregates of ectopically expressed, truncated ALS-associated TDP-43 mutants in Neuro2a cells contained M1-, K48-, and K63-ubiquitins in vitro, and the MG-132-mediated inhibition of proteasome activity enhanced the ubiquitination of TDP-43 (Figure 1, Figure 2 and Figure 3). These results indicate that TDP-43 aggregates include multiple types of ubiquitin linkages in Neuro2a cells, and may form branched and/or hybrid ubiquitin chains. Importantly, the genetic ablation of *Hoip* or treatment with the LUBAC inhibitor HOIPIN-8 drastically reduced the insoluble aggregates of truncated TDP-43 (Figure 4 and Figure 5). These results suggest that the conjugation of M1-ubiquitin to TDP-43 promotes the formation of insoluble aggregates. Indeed, M1-ubiquitin reportedly has a greater propensity to form fibrillar aggregates than K48- and K63-linked ubiquitin chains in vitro [36]. Interestingly, LUBAC seemed to be localized at the peripheries of the TDP-43 aggregates, and bound to TDP-43 in an E3 activity-dependent manner (Figure 6). We further detected the possible covalent linear ubiquitination of truncated TDP-43, and HOIPIN-8 suppressed the complex K48/K63/M1-ubiquitination of TDP-43. Since M1-ubiquitination is prohibited by N-terminally-tagged ubiquitin such as His_6_-tagged ubiquitin, the conjugation of M1-ubiquitin to TDP-43 may have been overlooked in previous in vitro mass spectrometry analyses. Thus, linear ubiquitin seems to play an indispensable role in the development of various neurodegenerative diseases through the promotion of liquid–liquid phase separation (LLPS), oligomerization, and aggregate formation. 

Currently, a glutamic neurotransmission inhibitor, riluzole, and an antioxidant drug, edaravone, are approved for ALS treatment [37]. Although these drugs prolong the survival of ALS patients, their efficacies are limited. Gene therapy clinical trials using antisense oligonucleotides (ASOs), RNA interference, adeno-associated virus (AAV)-mediated small hairpin RNA (shRNA), and antibody-based methods are currently being conducted [38]. Among them, the administration of ASOs targeting ataxin-2 reportedly reduced the TDP-43-positive aggregates and increased survival [39]. Thus, these gene therapies may be effective as disease-modifying treatments for ALS. In this study, we showed that aggregate-prone TDP-43 truncation mutants increased the TNF-α-mediated NF-κB activity, and this was suppressed by HOIPIN-8 (Figure 7), suggesting that LUBAC activity suppression by HOIPIN-8 or its derivatives will be a potential therapeutic target to suppress ALS-mediated neuroinflammation. Indeed, we have previously shown that HOIPIN-1 and -8 effectively suppressed activated B cell-like diffuse large B cell lymphoma (ABC-DLBCL) and psoriasis in mice models due to the inhibition of excessive NF-κB activity [26]. Our cellular analysis did not show a significant difference between WT^247-414^-GFP and A315T^247-414^-GFP, indicating that the production of the C-terminal fragment generally contributed to the aggregate formation, regardless of the single amino acid mutation of A315T. However, the human TDP-43 A315T mutation is a known etiology of ALS [40,41], and transgenic mice expressing the ALS-associated human A315T-mutant of TDP-43 reportedly exhibit ALS-like phenotypes with ubiquitin-positive inclusions [42,43]. Further in vivo investigations of the pharmacological and phenotypical effects of HOIPINs on ALS mice models are awaited. 

## Figures and Tables

**Figure 1 cells-11-02398-f001:**
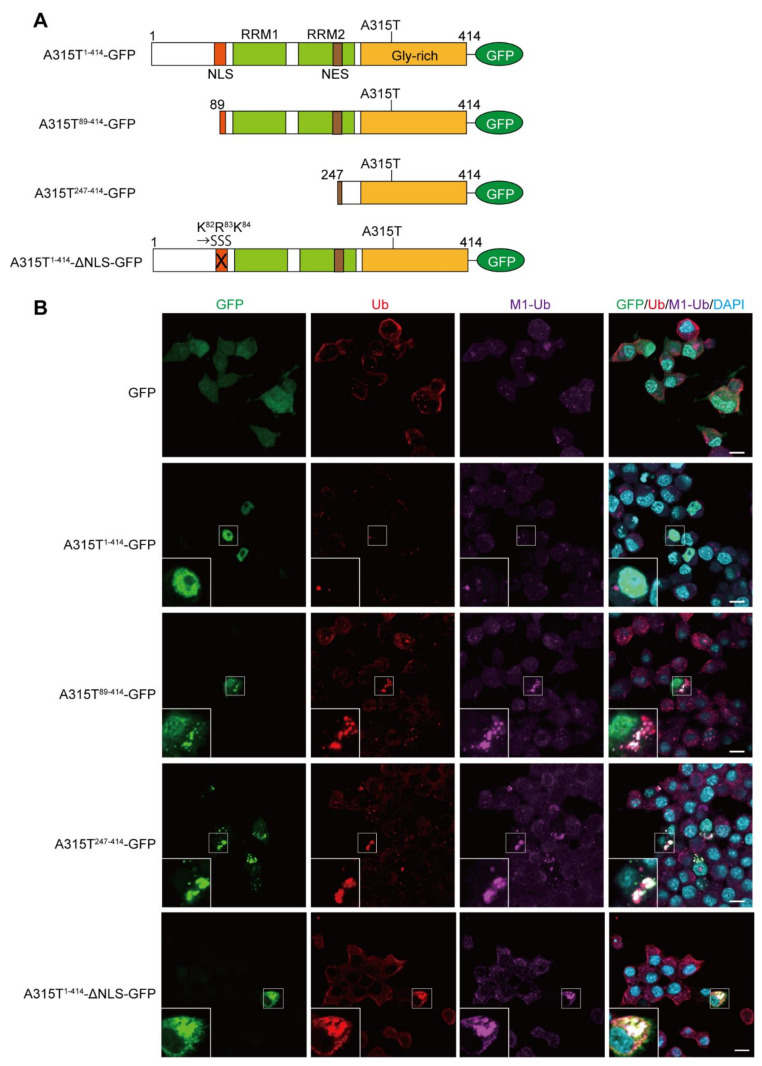
Linear ubiquitin is colocalized with cytoplasmic aggregates of truncated forms of A315T TDP-43. (**A**) Domain structures and constructed mutants of the C-terminally GFP-tagged TDP-43 with the ALS-associated A315T mutation are shown. NLS, nuclear localization signal; RRM—RNA recognition motif; NES—nuclear export signal. In A315T^1-414^-ΔNLS-GFP, the K^82^-R^83^-K^84^ amino acid sequence in NLS was replaced by S^82^-S^83^-S^84^. (**B**) Colocalization of linear ubiquitin with cytoplasmic aggregates of TDP-43. GFP and various A315T mutant of TDP-43-GFP were transiently expressed in Neuro2a cells, and the recruitments of pan-ubiquitin (Ub) and linear ubiquitin (M1-Ub) were visualized by immunofluorescence analyses. *Inserts*: Enlarged images of boxed regions. *Bars* = 10 μm.

**Figure 2 cells-11-02398-f002:**
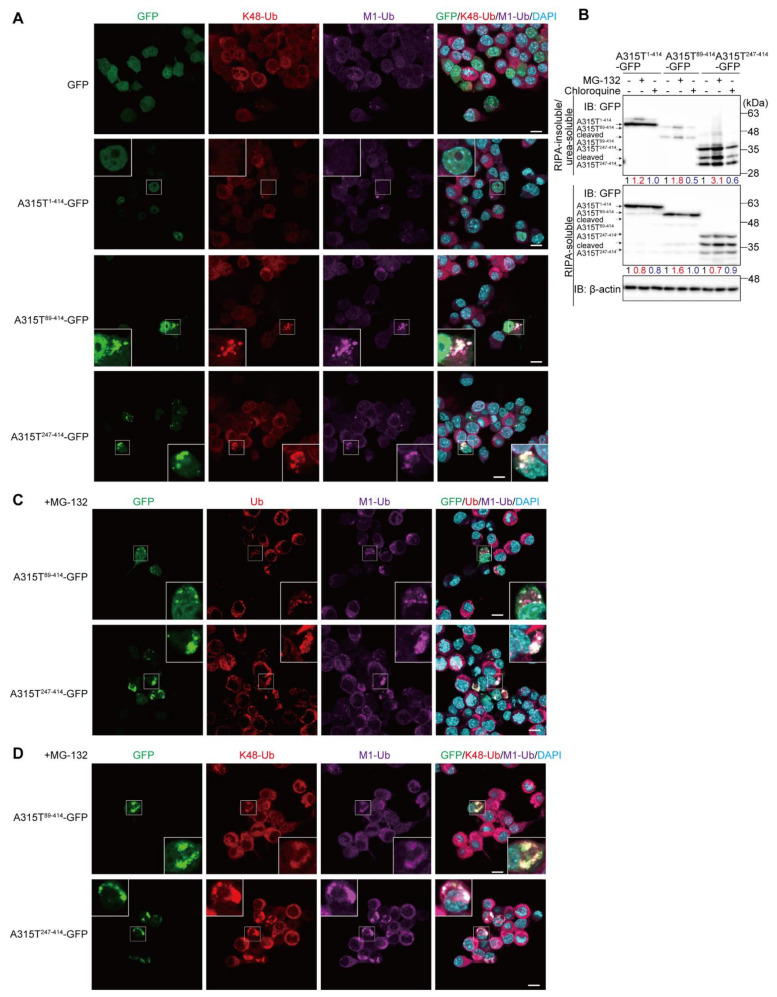
Cytoplasmic aggregates of truncated TDP-43 include M1- and K48-linked ubiquitin chains. (**A**) GFP and various A315T mutants of TDP-43-GFP were transiently expressed in Neuro2a cells, and the colocalization of K48-ubiquitin (K48-Ub) and linear ubiquitin (M1-Ub) was examined. (**B**) The suppression of proteasome activity increases insoluble A315T-GFPs. Plasmids encoding A315T^1-414^-GFP-, A315T^89-414^-GFP-, or A315T^247-414^-GFP were expressed in Neuro2a cells, which were treated with 2 μM MG-132 or 50 μM chloroquine for 6 h. The detergent-insoluble and 6 M urea-soluble fractions were immunoblotted with an anti-GFP antibody. Taking the intensity of the control as 1.0, the relative intensities are shown. (**C**,**D**) Enhanced aggregation of A315T-GFPs in the MG-132-treated Neuro2a cells. The colocalization of ALS-associated TDP-43 mutants with pan- and M1-ubiquitins (**C**), or K48- and M1-ubiquitins (**D**) was analyzed by immunofluorescent staining. (**A**,**C**,**D**) *Bars* = 10 μm.

**Figure 3 cells-11-02398-f003:**
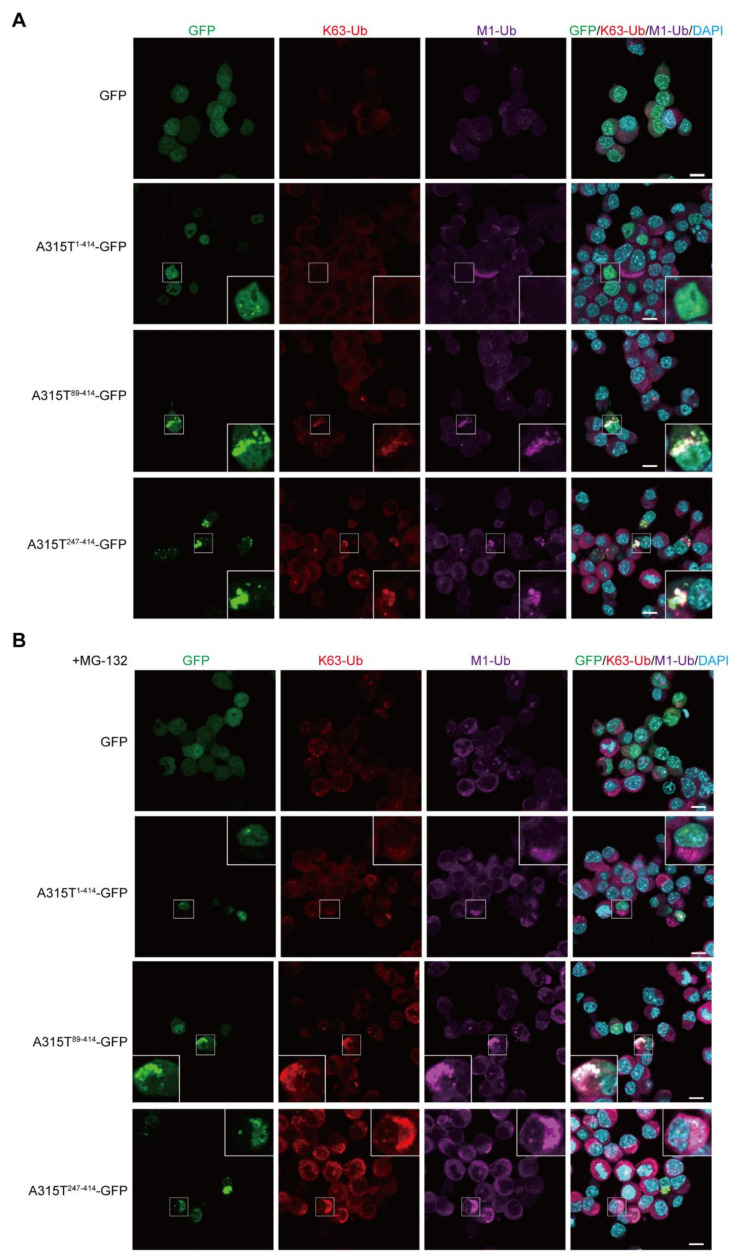
Cytoplasmic aggregates of truncated TDP-43 include K63- and M1-ubiquitins. (**A**) GFP and various A315T mutants of TDP-43-GFP were transiently expressed in Neuro2a cells, and the colocalization of K63-ubiquitin (K63-Ub) and linear ubiquitin (M1-Ub) was examined. (**B**) Similar experiments in (**A**) were performed after a treatment with 2 μM MG-132 for 6 h. *Bars* = 10 μm.

**Figure 4 cells-11-02398-f004:**
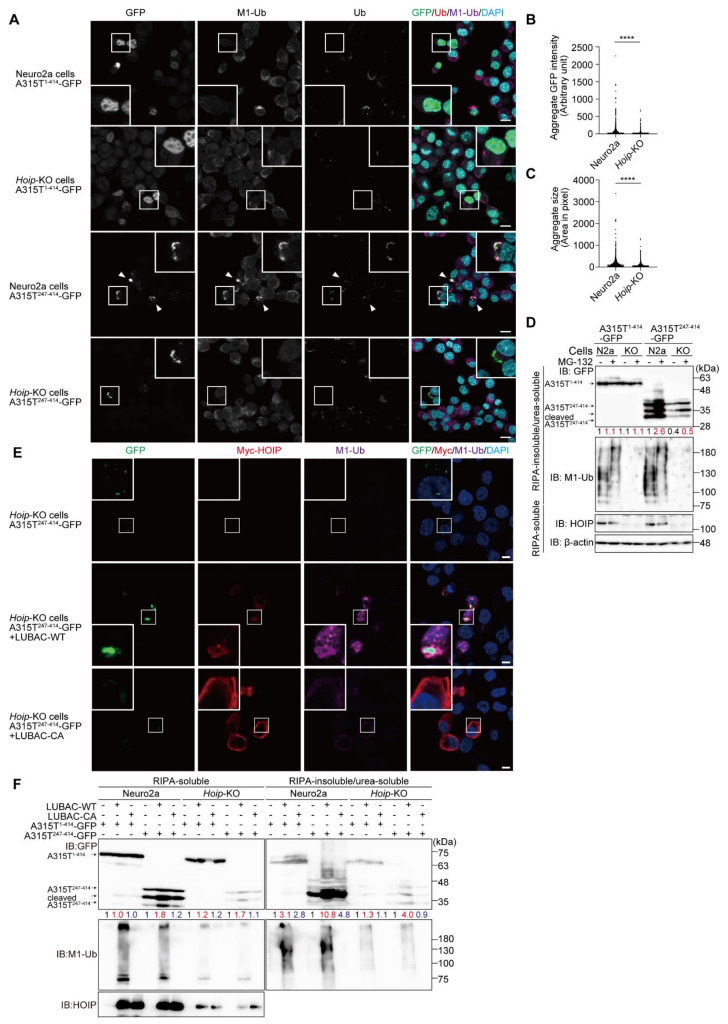
LUBAC-mediated linear ubiquitination activity is involved in cytoplasmic aggregation of the TDP-43 mutant. (**A**) Genetic ablation of *Hoip* reduces the cytoplasmic aggregation of truncated TDP-43. The full length and truncated mutants of TDP-43 (A315T^1-414^-GFP and A315T^247-414^-GFP) were expressed in the parental and *Hoip*-KO Neuro2a cells, and stained by the indicated immunofluorescent antibodies. *Bars* = 10 μm. (**B**,**C**) The intensities and sizes of the aggregates were reduced in the *Hoip*-KO cells. Aggregate GFP intensity (**B**) and aggregate size (**C**) were analyzed by Cell Profiler (Neuro2a, *n* = 2068; *Hoip*-KO, *n* = 2101 cells in 10 fields). Data are shown as scatter plots and were evaluated by the Mann–Whitney test. ****, *p* < 0.0001. (**D**) Reduced insoluble TDP-43 by the ablation of *Hoip*. A315T^1-414^-GFP and A315T^247-414^-GFP were expressed in the parental Neuro2a cells (N2a) and *Hoip*-deficient (KO) cells, and then treated with or without 2 μM MG-132. The RIPA-soluble and RIPA-insoluble/urea-soluble fractions were blotted by the indicated antibodies. Taking the intensity of the control as 1.0, the relative intensities are shown. (**E**) The restoration of LUBAC into the *Hoip*-KO cells recovers TDP-43 aggregates. A315T^247-414^-GFP was expressed in the *Hoip*-KO cells with wild-type (WT) or an active-site mutant of LUBAC. Immunofluorescent staining was then performed with the indicated antibodies. *Bars* = 5 μm. (**F**) The LUBAC-mediated linear ubiquitination activity facilitates TDP-43 aggregation. WT- and mutants of LUBAC and TDP-43 mutants (A315T^1-414^-GFP and A315T^247-414^-GFP) were expressed in the parental and *Hoip*-KO Neuro2a cells, as indicated. The RIPA-soluble and RIPA-insoluble/urea-soluble fractions were immunoblotted with the indicated antibodies. Taking the intensity of the control as 1.0, the relative intensities are shown.

**Figure 5 cells-11-02398-f005:**
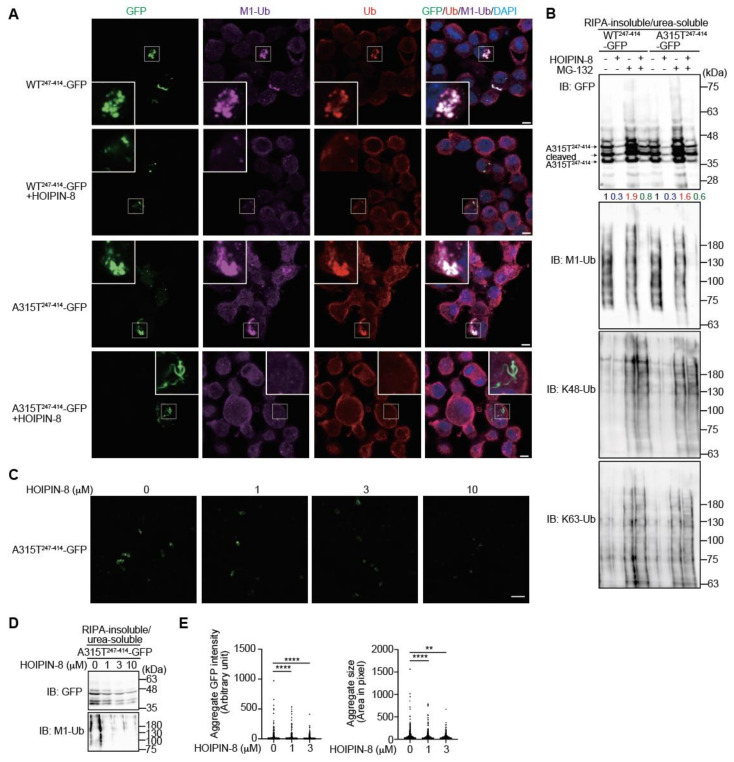
HOIPIN-8 ameliorates truncated TDP-43 aggregation. (**A**) Reduced aggregates of truncated TDP-43 in HOIPIN-8-treated cells. WT^247-414^-GFP or A315T^247-414^-GFP was expressed in Neuro2a cells in the absence or presence of 10 μM HOIPIN-8 for 20 h, and then immunofluorescent staining was performed with GFP or the indicated antibodies. *Bars* = 5 μm. (**B**) A HOIPIN-8-mediated decrease in the insoluble truncated TDP-43. Cells expressing WT^247-414^-GFP or A315T^247-414^-GFP were treated with 10 μM HOIPIN-8 and/or 2 μM MG-132, and the RIPA-insoluble/urea-soluble fractions were immunoblotted with the indicated antibodies. Taking the intensity of the control as 1.0, the relative intensities are shown. (**C**) Reduced A315T^247-414^-GFP aggregates by HOIPIN-8. A315T^247-414^-GFP were expressed in Neuro2a cells and treated with the indicated concentrations of HOIPIN-8 for 20 h. Then, the immunofluorescence of A315T^247-414^-GFP aggregates was analyzed. *Bar* = 40 μm. (**D**) Dose-dependent amelioration of insoluble A315T^247-414^-GFP by HOIPIN-8. The A315T^247-414^-GFP-expressing Neuro2a cells were treated with the indicated concentrations of HOIPIN-8 and 2 μM MG-132 for 6 h. The RIPA-insoluble/urea-soluble fractions were then subjected to immunoblotting using the indicated antibodies. (**E**) The intensities and sizes of the A315T^247-414^-GFP aggregates were reduced in the HOIPIN-8-treated A315T^247-414^-GFP-expressing Neuro2 cells. The aggregate GFP intensity and aggregate size in the A315T^247-414^-GFP-expressing Neuro2 cells shown in (**D**) were analyzed by Cell Profiler (0 μM, *n* = 2951; 1 μM, *n* = 3252; 3 μM, *n* = 2797 cells in 10 fields). Data are shown as scatter plots and were evaluated by the Kruskal–Wallis test. **: *p* < 0.01; ****: *p* < 0.0001.

**Figure 6 cells-11-02398-f006:**
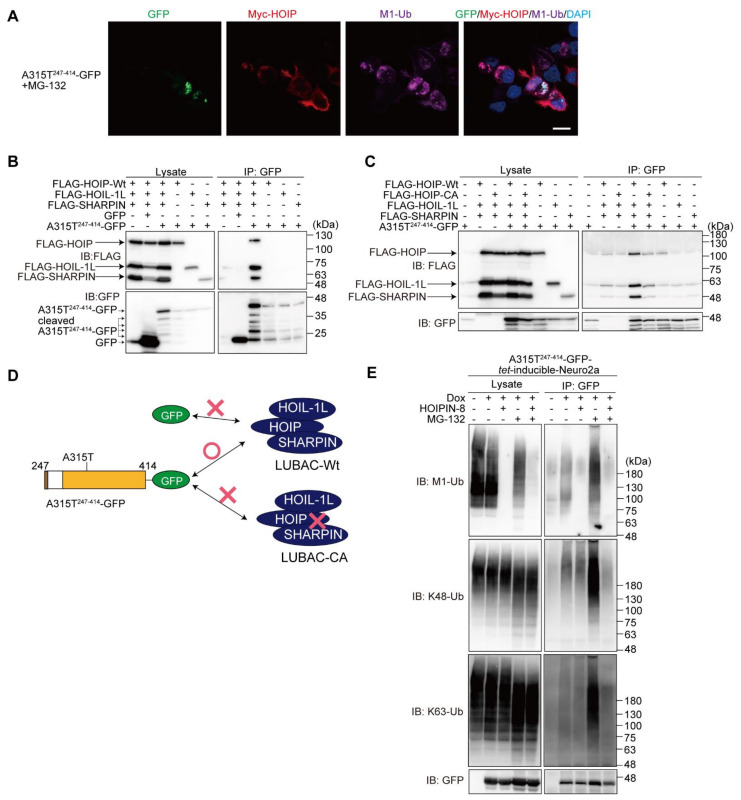
LUBAC activity is necessary for binding truncated TDP-43. (**A**) Colocalization of TDP-43 aggregates, LUBAC, and M1-ubiquitin. After the overexpression of LUBAC and A315T^247-414^-GFP in the presence of MG-132, immunofluorescence analyses of GFP, Myc-HOIP, and M1-ubiquitin were performed. *Bar* = 10 μm. (**B**) A315T^247-414^-GFP binds the LUBAC complex. FLAG-tagged LUBAC subunits, GFP, and/or A315T^247-414^-GFP were transfected into the HEK293T cells as indicated. Cell lysates and anti-GFP immunoprecipitates were immunoblotted with the indicated antibodies. (**C**) A315T^247-414^-GFP binds active LUBAC. A similar analysis as in (**B**) was performed, using WT or the active site C885A mutant of human HOIP, HOIP-CA. (**D**) A scheme of the truncated TDP-43-LUBAC binding. The results of (**B**,**C**) are summarized. (**E**) The effects of LUBAC and proteasome inhibitors on the M1-, K48-, and K63-ubiquitinations of A315T^247-414^-GFP. The A315T^247-414^-GFP-tet-inducible Neuro2a cells were treated with 1 μg/mL Dox and 10 μM HOIPIN-8 for 18 h, with 2 μM MG-132 added for the last 6 h, as indicated. Cell lysates obtained by the hot-SDS method and anti-GFP immunoprecipitates were immunoblotted with the indicated antibodies.

**Figure 7 cells-11-02398-f007:**
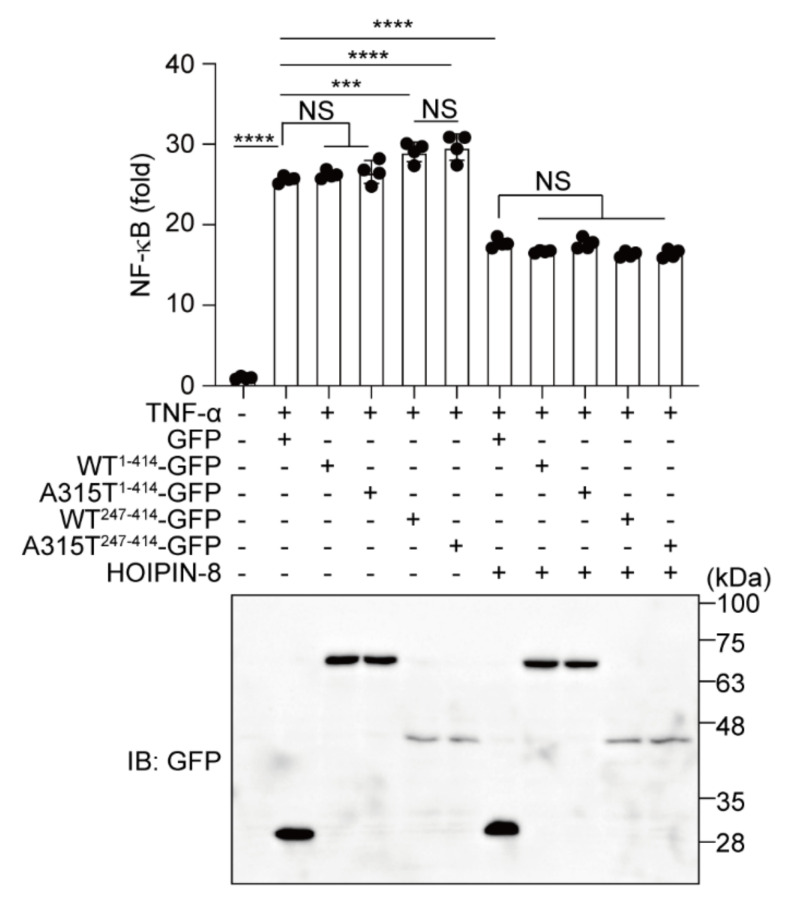
HOIPIN-8 suppresses the increased NF-κB activity by truncated TDP-43. Neuro2a cells transfected with the NF-κB luciferase reporter, WT^1-414^-GFP-, A315T^1-414^-GFP-, WT^247-414^-GFP, or A315T^247-414^-GFP with LUBAC were treated with 100 ng/mL TNF-α for 6 h, and as indicated, cells were pretreated with 10 μM HOIPIN-8 for 1 h before the TNF-α treatment. The luciferase activities were then analyzed. Data are shown as mean ±SEM, *n* = 4. One-way ANOVA was followed by a post hoc Tukey HSD test. ***: *p* < 0.001, ****: *p* < 0.0001, NS—not significant.

## Data Availability

The data presented in this study are available on request from the corresponding author.

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
