# Peer review of "Suppression of Linear Ubiquitination Ameliorates Cytoplasmic Aggregation of Truncated TDP-43"

_cells, 2022, doi:10.3390/cells11152398_

Round 1

Reviewer 1 Report

The authors Qiang et. al. in their manuscript titled “Suppression of linear ubiquitination ameliorates cytoplasmic aggregation of truncated TDP-43”, investigated effect of truncation on ubiquitination in TDP-43 protein. The authors have used in vitro model system to establish that truncation of TDP-43 impacts post-translation modification in terms of ubiquitin in TDP-43. The truncated TDP-43 with A315T mutation form cytoplasmic aggregates with multiple ubiquitin chains. Lastly, the authors suggest that genetic deletion of HOIP or inhibition of ubiquitination by LUBAC inhibitor, HOIPIN-8 suppresses cytoplasmic aggregation formation. This is an important study concerning cytoplasmic aggregation of mutated TDP-43 protein. There has not been a clear understanding about how mutated TDP-43 forms cytoplasmic aggregates and this study attempts to answer those question.

There are some questions that need to be answered to understand the manuscript in a better way:

1.       The authors show that truncation of TDP-43 favors cytoplasmic aggregations but not the full form of TDP-43A315T. Can the authors explain the reason behind this phenomenon?

2.       How the cytoplasmic aggregates affect cell? Which cellular structure/pathway these aggregates affect? Do they make cell degenerate? Do they send the cell in apoptotic route? Please explain these consequences.

3.       It will strengthen the authors hypothesis if they show using patient samples/mouse model tissue that TDP-43 truncation ends up showing similar results.

4.       Do the authors think truncation of TDP-43 could be used as a marker for disease?

5.       Have the authors checked that truncation of TDP-43 alters ubiquitination of proteins other than TDP-43. There are reports suggesting that cytoplasmic aggregates contain a large number of non TDP-43 ubiquitinated proteins.

6.       How does HOIPIN-8 treatment improve cellular function? Does the removal of cytoplasmic aggregates enough to make the cell healthier?       

Author Response

We greatly appreciate the reviewer’s advice and assistance, as their efforts have significantly improved the paper.

  1. The authors show that truncation of TDP-43 favors cytoplasmic aggregations but not the full form of TDP-43A315T. Can the authors explain the reason behind this phenomenon?

We thank the reviewer for this important comment. In most TDP-43 proteinopathies, TDP-43, including the A315T mutant, accumulates in the cytoplasm of neurons and is reduced in the nucleus, in a process strongly correlated with neurodegeneration (Barmada et al., 2010; Brettschneider et al., 2014; Ditsworth et al., 2017). The accumulation of cytoplasmic TDP-43 through overexpression of cytoplasmically-directed TDP-43 mutants in mice exhibited gain-of-toxicity in disease pathogenesis (Walker et al., 2015). In the cytoplasm, TDP-43 undergoes various post-translational modifications (PTMs), such as C-terminal fragmentation, phosphorylation, ubiquitination, acetylation, SUMOylation, nitrosylation, and methylation, and these PTMs affect the folding, droplet formation, and aggregation of TDP-43 (Francois-Moutal et al., 2019). The proteolytic cleavage of TDP-43 causes the generation of C-terminal fragments (CTFs) and facilitates the cytoplasmic accumulation of insoluble CTFs (Berning and Walker, 2019; Shenouda et al., 2022), since CTFs are intrinsically disordered low-complexity domains and form cross-β amyloid structures (Arseni et al., 2022).

We cited additional references in the Introduction and explained this phenomenon.

Arseni, D., Hasegawa, M., Murzin, A.G., Kametani, F., Arai, M., Yoshida, M., and Ryskeldi-Falcon, B. (2022). Structure of pathological TDP-43 filaments from ALS with FTLD. Nature 601, 139-143.

Barmada, S.J., Skibinski, G., Korb, E., Rao, E.J., Wu, J.Y., and Finkbeiner, S. (2010). Cytoplasmic mislocalization of TDP-43 is toxic to neurons and enhanced by a mutation associated with familial amyotrophic lateral sclerosis. J Neurosci 30, 639-649.

Berning, B.A., and Walker, A.K. (2019). The Pathobiology of TDP-43 C-Terminal Fragments in ALS and FTLD. Front Neurosci 13, 335.

Brettschneider, J., Arai, K., Del Tredici, K., Toledo, J.B., Robinson, J.L., Lee, E.B., Kuwabara, S., Shibuya, K., Irwin, D.J., Fang, L., et al. (2014). TDP-43 pathology and neuronal loss in amyotrophic lateral sclerosis spinal cord. Acta Neuropathol 128, 423-437.

Ditsworth, D., Maldonado, M., McAlonis-Downes, M., Sun, S., Seelman, A., Drenner, K., Arnold, E., Ling, S.C., Pizzo, D., Ravits, J., et al. (2017). Mutant TDP-43 within motor neurons drives disease onset but not progression in amyotrophic lateral sclerosis. Acta Neuropathol 133, 907-922.

Francois-Moutal, L., Perez-Miller, S., Scott, D.D., Miranda, V.G., Mollasalehi, N., and Khanna, M. (2019). Structural Insights Into TDP-43 and Effects of Post-translational Modifications. Front Mol Neurosci 12, 301.

Shenouda, M., Xiao, S., MacNair, L., Lau, A., and Robertson, J. (2022). A C-Terminally Truncated TDP-43 Splice Isoform Exhibits Neuronal Specific Cytoplasmic Aggregation and Contributes to TDP-43 Pathology in ALS. Front Neurosci 16, 868556.

Walker, A.K., Spiller, K.J., Ge, G., Zheng, A., Xu, Y., Zhou, M., Tripathy, K., Kwong, L.K., Trojanowski, J.Q., and Lee, V.M. (2015). Functional recovery in new mouse models of ALS/FTLD after clearance of pathological cytoplasmic TDP-43. Acta Neuropathol 130, 643-660.

Wang, X., Ma, M., Teng, J., Che, X., Zhang, W., Feng, S., Zhou, S., Zhang, Y., Wu, E., and Ding, X. (2015). Valproate Attenuates 25-kDa C-Terminal Fragment of TDP-43-Induced Neuronal Toxicity via Suppressing Endoplasmic Reticulum Stress and Activating Autophagy. Int J Biol Sci 11, 752-761.

  1. How the cytoplasmic aggregates affect cell? Which cellular structure/pathway these aggregates affect? Do they make cell degenerate? Do they send the cell in apoptotic route? Please explain these consequences.

We thank the reviewer for the comment. The overexpression of GFP-TDP-43-A315T in SH-SY5Y cells reportedly enhances apoptosis and autophagy (Wang et al., 2015). However, as shown in the new Figure 5SB, when we analyzed overexpression by using Neuro2a tet-on cells expressing empty vector, WT247-414-GFP, or A315T247-414-GFP with 24-h or 72-h induction by doxycycline, there was a slight decrease in cell proliferation after 72-h, although there was no statistically significant difference. Therefore, we did not perform further cell-level analyses of cell structural changes, cell denaturation, and apoptosis. In the manifestation of human pathology, the effects of chronic proteinopathy caused by TDP-43 aggregate formation cannot be ignored.

We added these results on page 11 in the manuscript.

Wang, X., Ma, M., Teng, J., Che, X., Zhang, W., Feng, S., Zhou, S., Zhang, Y., Wu, E., and Ding, X. (2015). Valproate Attenuates 25-kDa C-Terminal Fragment of TDP-43-Induced Neuronal Toxicity via Suppressing Endoplasmic Reticulum Stress and Activating Autophagy. Int J Biol Sci 11, 752-761.

  1. It will strengthen the authors hypothesis if they show using patient samples/mouse model tissue that TDP-43 truncation ends up showing similar results.

We would like to thank the reviewer for the crucial suggestions. We have already begun experiments to administer HOIPIN-8 to hTDP-43 A315T-transgenic (B6.Cg-tg(Prnp-TARDBP*A315T)95Balo/J) mice, which we obtained from Jackson Laboratory, although we don’t have patient samples. However, it takes some time to evaluate survival and motor function, and to perform pathohistochemical and cell biological analyses in vivo. It is an important research theme that we definitely want to clarify in the future. As a first step, we would like to show our results at the cellular level.

  1. Do the authors think truncation of TDP-43 could be used as a marker for disease?

We think that the truncation of TDP-43, which generates a low-complexity domain of CTFs, is important as a marker for diseases because it triggers a significant change in the physical characteristics of TDP-43.

  1. Have the authors checked that truncation of TDP-43 alters ubiquitination of proteins other than TDP-43. There are reports suggesting that cytoplasmic aggregates contain a large number of non TDP-43 ubiquitinated proteins.

We thank the reviewer for this important indication. It is well known that in ALS, various proteins such as FUS, OPTN, UBQLN2, C9ORF72, and SOD are included in aggregates, in addition to TDP-43. We believe that the ubiquitination status of proteins other than TDP-43 may change and affect cytoplasmic aggregates. However, most (~97%) of the ALS cases generate TDP-43-positive aggregates, so in this study we focused on TDP-43 expression and ubiquitination.

  1. How does HOIPIN-8 treatment improve cellular function? Does the removal of cytoplasmic aggregates enough to make the cell healthier?       

It is difficult to assess whether the cells have become healthier, as there was no significant difference in cell survival by the induction of TDP-43 aggregates, as shown in the new Figure 5SB. In order to address this point, we would like to analyze it in vivo, using ALS-model mice, in the future.

Reviewer 2 Report

Overall Comments

TAR DNA-binding protein 43 (TDP-43) is one of the major components of cytoplasmic inclusion in the brain and spines of amyotrophic lateral sclerosis (ALS). In their previous studies, authors reported the linear ubiquitin chain assembly complex (LUBAC) mediated Met1 (M1)-linked linear ubiquitin was colocalized with TDP-43 inclusion in neurons from familial and sporadic ALS patients and affected NF-κB activation and cell death. In order to test the effects of LUBAC mediated linear ubiquitination on the toxicity of TDP-43 aggregate, the authors conducted cell biological analyses with full length and nuclear localization signal missing truncated ALS associated Ala315Thr mutant of TDP-43 (A315T). Here, they found that the truncated forms of TDP-43 efficiently formed M1-, Lys (K)48, K63-positive cytoplasmic inclusions and that HOIP, a key component of LUBAC, KO and HOIPIN-8, a LUBAC inhibitor suppressed cytoplasmic aggregates and NF-κB activation. Their results suggested that multiple ubiquitination of TDP-43 influence protein aggregation and inflammatory responses and that a LUBAC pathway is a target for intervention. They express diverse forms of proteins in cells and present many cell-biological data with to claim their idea. There is the lack of quantitative analysis, clear presentation, and explanation of their figures for better understating and supporting.

Major issues:

1.     In most of immunostaining figures, authors focused a single inclusion positive cell to show the influence of the mutants and treatment. It is not easy but they may need mention how many cells among the total cells behave similarly.

2.     In Figure 2B and other WB results, it is difficult to understand the results because multiple bands show up and the information on the protein size and identy is missing. Authors said that MG-132 treatment increased insoluble A315T-GFP but they may need statistical analysis to support their mention. The statistical analysis may also need in other WB figures to persuade the readers

3.     In Figure 3, authors’ suggestion (line 253-257) came from immunostaining with a single cell.  As mentioned above, additional data may need support their idea.

4.     Figure 4F, 5B, 6B, and 6C contain many information and lanes. They may need find other way to present their results to make easy to follow.

5.     As presented on Figure 5A and Figure 7, the wild type TDP-43 and the mutant A315T-GFP did not show any difference in their effects and it seems that the LUBAC mediated cytoplasmic aggregation is independent of the mutation. Because many result came from the mutant form of TDP-43, author may need to explain why they use A315T TDP-43 and their interpretation.

6.     Overall, they need present the immunostaining and the WB results clearly to improve their claims.

Author Response

We greatly appreciate the reviewer’s advice and assistance, which significantly improved the paper. The English text of this manuscript has been reviewed by a native English researcher.

  1. In most of immunostaining figures, authors focused a single inclusion positive cell to show the influence of the mutants and treatment. It is not easy but they may need mention how many cells among the total cells behave similarly.

We thank the reviewer for this important comment. In transient expression, the plasmid is not introduced into all cells, so the number of aggregate-positive cells in the immunofluorescence staining field of view will unavoidably decrease. Of course, we have seen multiple fields of view and confirmed that they have similar trends, and we showed representatives to prevent redundancy. In addition, we showed the quantification (Figs. 4B, 4C, and 5E) and colocalization (Figs. S1 and S2) of immunofluorescence staining, and biochemical immunoblotting analyses showing the behavior of the entire cell. Therefore, we believe that our cell-level analysis is sufficient.

  1. In Figure 2B and other WB results, it is difficult to understand the results because multiple bands show up and the information on the protein size and identy is missing. Authors said that MG-132 treatment increased insoluble A315T-GFP but they may need statistical analysis to support their mention. The statistical analysis may also need in other WB figures to persuade the readers

Regarding protein sizes, we think that it is common to display the migrations of molecular weight standards in a WB. To respond to the reviewer's suggestions, we modified Figs. 2B, 4D, 4F, 5B, and 6B to indicate the sources of the proteins. Furthermore, in Figs. 2B, 4D, 4F, and 5B, the intensities of the corresponding bands were quantified, and in comparison with the controls, the relative intensities are shown.

  1. In Figure 3, authors’ suggestion (line 253-257) came from immunostaining with a single cell. As mentioned above, additional data may need support their idea.

To demonstrate multiple ubiquitinations on the whole cell level, we now present immunoblots by the anti-K63-Ub antibody in the new Fig. 6E. The anti-GFP immunoprecipitation analysis using hot-SDS lysates (right panels) clearly indicated that M1-, K48-, and K63-ubiquitin chains are covalently conjugated to A315T247-414-GFP (lane 1 vs. 2) and the inhibition of proteasome activity by MG-132 enhances multiple ubiquitinations (lane 4). Although HOIPIN-8 extensively removes M1-ubiquitination (lane 3), it has a suppressive effect on multiple ubiquitination (lane 5). Therefore, we believe the suggestion of multiple ubiquitinations is correct.

  1. Figure 4F, 5B, 6B, and 6C contain many information and lanes. They may need find other way to present their results to make easy to follow.

It is difficult to reduce lanes for comparison with the control, but as mentioned above, we indicated the migration of proteins and included the quantification of the band intensities in Figs. 4F and 5B. Moreover, a scheme has been added as the new Fig. 6D to summarize the results of Fig. 6B and 6C.

  1. As presented on Figure 5A and Figure 7, the wild type TDP-43 and the mutant A315T-GFP did not show any difference in their effects and it seems that the LUBAC mediated cytoplasmic aggregation is independent of the mutation. Because many result came from the mutant form of TDP-43, author may need to explain why they use A315T TDP-43 and their interpretation.

As the reviewer pointed out, the cellular analysis did not show a significant difference between WT247-414-GFP and A315T247-414-GFP, indicating that the production of the C-terminal fragment, rather than the fragment with the single amino acid replacement of Ala315 by Thr, contributed to aggregate formation. However, the human TDP-43 A315T mutation has many research achievements as an etiology of ALS, and model mice (B6.Cg-tg(Prnp-TARDBP*A315T)95Balo/J) that form ubiquitin-positive inclusions and die after motor dysfunction are also available. Therefore, we believe that it is useful to proceed with our basic cellular analysis using the A315T mutant for comparisons of in vitro and in vivo results. We added this in the Discussion.

  1. Overall, they need present the immunostaining and the WB results clearly to improve their claims.

We have tried to improve our figures, as mentioned above.

Round 2

Reviewer 2 Report

Overall Comments:

After the authors tried to answer and to reflect the comments, their manuscript seems to be

better for me to understand although several questions and minor issues still remain to be

revised. I still feel the lack of more detailed analysis of their western blot results but

considering the other results, their results and interpretation look reasonable. After they

answer following questions, I think this manuscript is qualified for publication in Cells.

Issues:

1. On page 4. The abbreviation of “The UBAN domain” appears first. The authors need

to write full name and may explain the word.

2. On page 11, in the ‘Figure 1’ legend, “mutants of TDP-43 mutants” may need to be

revised like “truncated forms of A315T TDP-43”.

3. On page 12, when they explained ‘Figure 2B’, the authors mentioned “the amounts of

full length and truncated forms of A315T-GFP in the insoluble aggregates fraction

were in the presence of MG-132, but not chloroquine”. Compared with the truncated

forms, the full length form of A315T-GFP increased by much less extents. Without

more detailed analysis, their interpretation may not be acceptable. In addition, upon

their analysis, the chloroquine treatment reduced insoluble aggregates almost 50%.

Please explain whether this meaningful and whether autophagy inhibition is able to

inhibit aggregation formation.

4. On page 15, when they explain ‘Figure 3B’, the author said “in the presence of MG-

132, the cytoplasmic aggregates of A315T89-414-GFP and A315T247-414-GFP

colocalized with K63- and M1-ubiquitins were increased’. But without direct

comparison as shown in the Figure 2B, their claim is not fully convinced although the

cytoplasmic aggregates seem to increase in the Figure 3B.

Typo and minors:

1. In second lane on page 4, acetylayion is to be acetylation

2. In first lane on page 23, “both” may need to be removed.

3. In second lane on page 23, “not the K-48 and K63-ubiquitination” may be to be “not

the K-48 nor K63-ubiquitination”.

4. In 6 th lane on page 25 in Discussion, “oligomerized with a prion-like filamentous

structure” may be revised as ‘is oligomerized with a prion-like filamentous structure’

Evaluation:

This manuscript still needs some revisions but is qualified for publication in Cells after

revised.
